# Trauma-Induced Coagulopathy: A Review of Specific Molecular Mechanisms

**DOI:** 10.3390/diagnostics15111435

**Published:** 2025-06-05

**Authors:** Andrea Capponi, Carlo Rostagno

**Affiliations:** 1Dipartimento Medicina Sperimentale e Clinica, Università di Firenze, 50134 Firenze, Italy; andrea.capponi@unifi.it; 2Dipartimento Medicina Sperimentale e Clinica, Medicina Interna 3 AOU Careggi, Università di Firenze, Viale Morgagni 85, 50134 Firenze, Italy

**Keywords:** trauma, coagulation, bleeding, thrombomodulin, fibrinolysis

## Abstract

Trauma remains a leading cause of death and disability in adults, and about 20% of deaths occur due to intractable bleeding. Trauma-induced coagulopathy (TIC) is a complex hemostatic disorder characterized by an abnormal coagulation response, which can manifest as either a hypo-coagulable state, leading to excessive bleeding, or a hypercoagulable state, resulting in thromboembolic events and multiple organ failure. Early diagnosis and correction of hypocoagulability may be lifesaving. Replacement of coagulation factors using blood components as well as counteracting enhanced fibrinolysis with tranexamic acid in association with a strategy of damage control are the current practices in the management of TIC. Nevertheless, the improved comprehension of the several mechanisms involved in the development of TIC might offer space for a tailored treatment with improvement of clinical outcome. This review aims to outline the pathophysiology of TIC and evaluate both established and emerging management strategies. A thorough literature review was made with a specific emphasis on articles discussing the molecular mechanisms of trauma-induced coagulopathy. We utilized PubMed, Scopus, and Web of Science with the main search terms “trauma-induced coagulopathy”, “molecular mechanisms”, and “coagulation pathways”.

## 1. Introduction

Despite advances in resuscitation, surgical management, and critical care, trauma remains a leading cause of death and disability in adults [1]. Bleeding is the more frequent cause of death in severe trauma following head injury and is related to two main mechanisms: fatal hemorrhage from vascular injury and bleeding secondary to coagulopathy [2,3]. Coagulopathy can result from physiological disturbances such as acidosis, hypothermia, or hemodilution due to fluid or blood administration. However, an acute coagulopathy can occur in severely injured patients independently from or in addition to these factors. Traumatic coagulopathy represents a unique pathological entity that in time has been called in various ways: trauma-induced coagulopathy (TIC), acute traumatic coagulopathy (ATC), early coagulopathy of trauma (ECT), and acute coagulopathy of trauma-shock (ACoTS) [4]. The severity of TIC strongly correlated with the combined degree of both injury and shock. It is a complex hemostatic disorder characterized by an abnormal coagulation response, which can manifest as either a hypo-coagulable state, leading to excessive bleeding, or a hypercoagulable state, resulting in thromboembolic events and multiple organ failure [4,5,6].

Although improved efficiency in military and civilian trauma systems has shortened the time between acute injury and treatment, between 25 and 35 percent of injured civilians exhibit biochemically evident coagulopathy upon arrival at the emergency department [2,7,8,9,10], resulting in fatality in 30–50% of cases. Patients who develop TIC receive on average a higher number of transfusions, spend more time in intensive care, undergo more days on mechanical ventilation, and have a higher incidence of multi-organ dysfunction. Most deaths in trauma patients related to in-hospital hemorrhage occur within six hours of admission [11,12]. The pathophysiology of TIC involves several mechanisms, including tissue injury, shock, endothelial dysfunction, platelet activation, and dysregulated fibrinolysis. Early TIC is often associated with hypocoagulability due to factors such as fibrinogen depletion, impaired thrombin generation, and platelet dysfunction, and is exacerbated by the “lethal triad” of coagulopathy, hypothermia, and acidosis [13,14]. In contrast, late TIC can present as a hypercoagulable state, driven by ongoing inflammation, endothelial injury, and a shift towards a prothrombotic phenotype. This phase is often linked to venous thromboembolism and multiple organ failure.

## 2. Methods

To identify relevant studies, a comprehensive literature search was conducted in different databases. We searched for relevant articles published between 2000 and 2025 in the three databases PubMed/MEDLINE, Scopus, and Web of Science. All articles that meet the following key criteria were selected: trauma-induced coagulopathy”, “molecular mechanisms”, and “coagulation pathways”.

The selection of articles was subject to a consistent review and assessment in order to identify studies that were potentially relevant to the objectives of this review.

The main inclusion criteria were as follows: (1) articles in English, (2) original studies investigating trauma, and (3) studies examining the role of different mechanisms in bleeding after trauma.

This review excluded editorials, case reports, and letters. Studies that met the inclusion criteria were further analyzed, and relevant data were extracted and evaluated for each article. Any discordances between investigators were resolved through a consensus approach.

## 3. Epidemiology and Early Detection of TIC

TIC occurs in about 25% of severely injured patients, and its incidence is directly proportional to the severity of the injury, defined by the injury severity score (ISS) [15]. The predominant cause of death after trauma continues to be central nervous system (CNS) injury (21.6–71.5%), followed by exsanguination (12.5–26.6%), while sepsis (3.1–17%) and multi-organ failure (MOF) (1.6–9%) are the predominant causes of late death [16]. Bleeding-related worsening of intracranial injuries in cases of brain trauma may be ominous [16]. Patients with coagulopathy (TIC+) are more severely injured [ISS—average ISS score 34 vs. 25, *p* < 0.001], have more severe shock (systolic blood pressure 101.5 ± 33.8 mmHg vs. 110.4 ± 29.9 mmHg, *p* < 0.001), higher substantial bleeding (69.2 vs. 27%, *p* < 0.0001), higher preponderance of multi-organ failure (3.7% vs. 1.0, *p* < 0.01), and a significantly higher in-hospital mortality rate (52.3% vs. 12.4%, *p* < 0.001), in comparison to non-coagulopathic patients (TIC−) [17].

The diagnosis of TIC involves both conventional coagulation tests and viscoelastic hemostatic assays. Conventional tests include: platelet count, Clauss assay, international normalized ratio (INR), thrombin time (TT), prothrombin time (PT), and activated partial thromboplastin time (aPTT).

Viscoelastic hemostatic assays, such as thromboelastography (TEG) and rotational thromboelastometry (ROTEM), provide dynamic assessments of clot formation and stability, offering more detailed insights into the coagulopathic state. These assays are particularly useful for identifying hyperfibrinolysis, a critical component of TIC. Additionally, clinical scoring systems like the Trauma-Induced Coagulopathy Clinical Score (TIC Score) have been developed to aid in the early detection of TIC. These scores incorporate parameters such as Glasgow Coma Scale, Shock Index, hemoglobin levels, prehospital fluid volume, and the use of norepinephrine.

The timing of diagnostic tests significantly affects the accuracy of diagnosing TIC using both conventional coagulation tests and viscoelastic hemostatic assays.

### 3.1. Conventional Coagulation Tests

The time required for laboratory processing decreases the clinical relevance of conventional hemostasis laboratory tests. A median time of 78 min from blood collection to PT results has been reported, which can be too slow for timely clinical decision-making in acute trauma settings, sometimes leading to underestimation of the severity of coagulopathy and inappropriate management [18]. Although INR has been commonly used as a trauma-related indicator of coagulopathy, INR was designed and intended only for patients on oral anticoagulant therapy for monitoring oral anticoagulant therapy across different labs and not for screening of coagulation disorders. A prolongation of the prothrombin time ratio and activated partial thromboplastin time (aPTT) is usually detected in shocked patients, defined as an admission base deficit of greater than 6 mmol/L. According to PROMMTT study criteria, an aPTT ≥ 35 s clearly defined a condition of hypocoagulability [19].

The use of traditional clotting tests has limited sensitivity in TIC diagnosis; however, when abnormal, a higher mortality rate has been reported [20].

### 3.2. Viscoelastic Hemostatic Assays

Thromboelastography (TEG) and rotational thromboelastometry (ROTEM): These assays provide rapid and dynamic assessment of coagulation status. For instance, ROTEM can identify acute traumatic coagulopathy within 5 min using clot amplitude measurements, being significantly faster than conventional tests. Early use of viscoelastic assays can guide more timely and appropriate transfusion strategies, reducing the risk of both bleeding and thromboembolic complications.

The rapid availability of results from viscoelastic assays makes them particularly valuable in the acute phase of trauma care. Meizoso et al. highlighted that while viscoelastic hemostatic assays are most beneficial when used early, current technology limits their use in pre-hospital settings, and their utility in the emergency department can be compromised by the time to results [21]. However, their application in the operating room and intensive care unit has shown compelling data supporting their value in guiding hemostatic therapy [18].

There is not a commonly accepted visco-elastic definition of ATC. The main variables retrieved from EXTEM ROTEM are clotting time [CT (s)]; clot formation time [CFT (s)]; A10 clot amplitude 10 min after the end of (CT); maximum clot firmness [MCF (mm)]; and maximum lysis time at 30 min [ML30 (%)]. For the FIBTEM assay, A10 and MCF. Increases in clotting time and clot formation time and loss of clot amplitude (CA) and maximal clot amplitude are commonly associated with hypocoagulability. Several algorithms were created to guide factor replacement according to different ROTEM patterns of hypocoagulability, although none was fully validated by randomized trials [21].

In summary, the timing of diagnostic tests is crucial for the accurate diagnosis of TIC. Viscoelastic hemostatic assays like TEG and ROTEM offer faster and more detailed assessments compared to conventional coagulation tests, thereby improving the timing and accuracy of TIC diagnosis and management.

## 4. Physiopathological Mechanisms

### 4.1. Lethal Triad: An Outdated Concept

Historically, TIC has been attributed to the consumption of coagulative proteases due to hemodilution from resuscitation, hypothermia, and acidosis—collectively known as the “lethal triad”. However, recent research suggests that none of these factors alone was the cause of TIC. Instead, the primary trigger appears to be shock status and systemic hypoperfusion [22].

Tissues with low perfusion pressures typically develop an anticoagulant environment through the activation of protein C and its cascade, which prevents thrombotic events at the vascular level. In severe trauma with shock conditions, this physiological response is exacerbated, leading to pathological hemorrhagic tendencies. The precise extent of activation of this pathway in vivo remains to be fully elucidated, but it appears to be influenced significantly by the severity of trauma and subsequent systemic hypoperfusion.

Acidosis contributes to clotting dysfunction by disrupting the assembly of coagulation factor complexes involving calcium and phospholipids, particularly at pH below 7.2. Correcting acidosis alone, however, does not always reverse associated coagulopathy, suggesting that tissue injury contributes to coagulopathy through additional mechanisms [23,24]. Hypothermia, induced by exposure to cold during injury and transport, as well as through administration of cold fluids, further complicates the coagulation profile. Surgical patients are particularly vulnerable to hypothermia due to prolonged exposure in the operating room, additional fluid administration, and the effects of anesthesia. However, hypothermia itself is not a strong independent predictor of mortality [25,26].

Resuscitation-associated coagulopathy (RAC), also known as iatrogenic coagulopathy, refers to disruptions in the coagulation system caused by large volumes of intravenous fluids or unbalanced blood product administration during shock management [27,28]. The age of blood products may also contribute to RAC.

In summary, TIC represents a disturbance in hemostasis and activation of fibrinolysis that occurs early after injury (Figure 1), often manifesting biochemically before the development of significant acidosis, hypothermia, or hemodilution. Risk factors for TIC include hypotension, higher injury severity scores, worsening base deficit, and head injury [10,29,30]. Once established, TIC can be compounded by other causes of coagulopathy.

### 4.2. TIC and DIC: Sibling Coagulopathies

Coagulopathy observed in TIC, in the absence of thrombocytopenia and hypofibrinogenemia, suggests that consumption alone may not be the primary underlying mechanism. In acutely injured patients, elevated D-dimer levels and depleted fibrinogen levels indicate intravascular fibrin deposition and active fibrinolysis [31,32]. However, studies have shown that functional thrombin generation, assessed through the presence of prothrombin fragments and thrombin–antithrombin complexes, remains intact in these patients [22,33,34].

Thrombin modulates immune responses because thrombin can clear protease-activated receptors on endothelial cells, immune cells, and platelets, leading to the release of cytokines, chemokines, and adhesion molecules [35].

TIC typically occurs when tissue injury is coupled with systemic hypoperfusion, suggesting that the mechanism behind TIC is likely distinct from disseminated intravascular coagulation (DIC), although these conditions frequently overlap. Further research is needed to better understand these distinctions and their implications for clinical management.

### 4.3. Hyperfibrinolysis and Fibrinolysis Shutdown

The pathophysiology of TIC involves several complex mechanisms that lead to both hyperfibrinolysis and fibrinolysis shutdown. These processes are primarily driven by the effects of shock, hypoperfusion, and direct tissue injury on the endothelium.

#### 4.3.1. Hyperfibrinolysis

The cleavage of fibrinogen into fibrin and fibrin polymerization are necessary for the stabilization of blood clots. Decreased fibrinogen concentrations are associated with an increased transfusion rate and mortality in trauma patients [36]. Otherwise, plasmin degradation of fibrin is essential for maintaining vessel patency. An excessive activation of the system, hyperfibrinolysis, is associated with the risk of intractable bleeding after injury [37]. Hyperfibrinolysis in TIC is triggered by endothelial thrombomodulin expression upregulation in response to tissue hypoperfusion. Thrombomodulin forms a complex with thrombin generated by tissue trauma, accelerating the activation of protein C. Activated protein C then contributes to coagulopathy by inactivating factors Va and VIIIa and promotes fibrinolysis by inhibiting plasminogen activator inhibitor 1 (PAI-1). Moreover hypoxemia and adrenergic activation activate endothelial cells, and significant amounts of tissue plasminogen activator (tPA) are released from the Weibel–Palade vesicles into the bloodstream. This hypothesis is supported by the finding that hyperfibrinolysis has also been demonstrated in other nontraumatic low-flow states, such as life-threatening anaphylactic shock or out-of-hospital cardiac arrest [38,39].

Endothelial tPA overexpression is crucial for hyperfibrinolysis, as tPA forms an inactive complex with PAI-1, reducing its activity. This phenomenon was validated using TEG with exogenous tPA challenge, highlighting the role of severe hypoperfusion in releasing tPA into circulation and sequestering PAI-1 [40].

#### 4.3.2. Fibrinolysis Shutdown

Conversely, fibrinolysis shutdown in TIC is characterized by impaired release of tPA following trauma. This condition is associated with higher mortality compared to physiologic fibrinolysis. Moore et al. [41] categorized patients with severe trauma based on clot lysis characteristics: fibrinolysis shutdown, physiologic lysis, and hyperfibrinolysis. Mortality rates were significantly elevated in patients with fibrinolysis shutdown and hyperfibrinolysis, (respectively, 22 and 37%), with deaths from massive hemorrhage in hyperfibrinolysis and later deaths from multi-organ failure attributed to microcirculatory fibrin deposits in fibrinolysis shutdown [42].

These distinct fibrinolytic phenotypes underscore the complexity of TIC and highlight the critical importance of timely and targeted interventions to mitigate coagulopathic complications in trauma patients.

### 4.4. The New Concept: Trauma Endotheliopathy (EoT)

In physiological conditions, endothelium expresses many anticoagulant molecules, including thrombomodulin, endothelium PC receptors, and endothelium glycocalyx layer complexes (EGL). ECL is a matrix of proteoglycans (syndecan-1, hyaluronic acid, and heparan sulfate) on the surface of endothelial cells. Loss of EGL (shedding) has been detected in a variety of inflammatory conditions, including trauma. These conditions induced by various molecular mechanisms that contribute to endothelial cell injury, activation, and maladaptive responses following major injuries have been termed shock-induced endotheliopathy (SHINE) or endotheliopathy of trauma (EoT) [43]. Three main types of endotheliopathy: (a) damage/loss of the endothelial glycocalyx, (b) cleavage of soluble thrombomodulin (sTM) with impairment of the natural protein C anticoagulant system, and finally (c) increased permeability due to loss of the integrity of endothelial intercellular junctions have been described. Laboratory studies showed that inhibition of catecholamine secretion significantly decreased expression of markers of endothelial injury.

#### 4.4.1. Endothelial Damage

Reduced perfusion and tissue damage induce exposition of a greater number of thrombomodulin molecules on the endothelial surface, with the formation of thrombin-thrombomodulin complexes. The latter induce the activation of large amounts of protein C, which in turn binds factors V and VIII. The consumption of factors, as well as of thrombin and thrombomodulin, causes a reduction in the rate of activation of the coagulation cascade and a lower stability of the formed clot (Figure 2) [44].

Compromission of endothelial glycocalyx, a protective layer on the endothelial surface, is observed not rarely after trauma. This disruption increases vascular permeability, contributing to coagulopathy and inflammation [45]. Syndecan-1, a glycocalyx degradation product, is released in response to glycocalyx damage, and plasma concentrations correlate with coagulopathy and mortality [46,47].

Shedding of endogenous heparan sulfates from the glycocalyx can lead to auto-anticoagulation via increased circulating endogenous heparinoids. The extent of glycocalyx damage correlates with catecholamine levels post-injury [48]. Elevated syndecan-1 levels upon admission are predictive of increased mortality and the need for blood transfusions. For example, levels ≥ 40 ng/mL have been linked to higher 30-day in-hospital mortality rates [46]. High syndecan-1 levels are associated with elevated inflammatory markers (such as IL-6 and IL-10) and coagulation markers (including D-dimer and tissue plasminogen activator [tPA]). These associations underscore the marker’s role in reflecting the extent of inflammation and coagulopathy in trauma patients. In pediatric trauma cases, elevated syndecan-1 levels are associated with shock and poorer outcomes [49]. Each 10 ng/mL increase in syndecan-1 correlates with 10% higher odds of death or requiring transfusion [50].

Syndecan-1 levels are also implicated in the development of DIC following trauma. Studies have shown that elevated syndecan-1 correlates with intense and prolonged activation of coagulation pathways, contributing to DIC progression [51]. In summary, syndecan-1 may be a useful marker of endothelial injury severity in trauma patients. Its elevation signifies worse clinical outcomes, heightened inflammation, increased coagulopathy, and a heightened risk of developing DIC. Monitoring syndecan-1 levels can aid in early intervention and management strategies to mitigate these complications in trauma care.

#### 4.4.2. Von Willebrand Factor (VWF) and ADAMTS13

Release of VWF by endothelial cells occurs approximately rapidly after thrombin activation. In an early phase of trauma, VWF levels in patients with coagulopathy were lower than those in patients without coagulopathy. The admission VWF levels were also correlated with the admission PC/FVII levels, implying that early low VWF levels might be mainly attributed to coagulopathy probably related to an impairment of thrombin generation. Nevertheless, in surviving patients VWF peaked in the week after admission [52]. Release of hyperadhesive, ultra-large VWF multimers from endothelial cells may contribute to the hypercoagulable phase of TIC, leading to microvascular thrombosis. Concurrently, a deficiency in ADAMTS13, the VWF-cleaving metalloprotease, exacerbates this condition, contributing to coagulopathy and endothelial dysfunction [53]. When analyzing by ISS, patients with ISS > 15 had lower ADAMTS13 activity. After multivariable linear regression, ADAMTS activity was independently associated with coagulopathy [54]. The imbalance between ADAMTS13 activity and levels of VWF results in the persistence of ultra-large VWF molecules circulating in the bloodstream. and is associated with more severe coagulation abnormalities, greater blood loss, and increased need for blood transfusions in trauma patients [55].

#### 4.4.3. Calcium Influx

Endothelial permeability induced by trauma involves calcium influx through the TRPV4 channel. This influx leads to myosin light chain phosphorylation and actomyosin contraction, disrupting endothelial junctions and increasing permeability [56]. TRPV4 is highly expressed on vascular endothelial cells and can be activated by many stimuli, including inflammatory mediators, reactive oxygen species, and acidosis, all potential contributors to a pathologic milieu after major trauma. Pharmacologic inhibition of TRPV4 restores endothelial function and improves survival in multiple murine models of sepsis. Conversely, TRPV4 agonism results in massive endothelial barrier dysfunction with profound circulatory collapse, resembling the systemic inflammatory response phenotype frequently observed in critically injured patients [57]. Ex vivo plasma from injured patients with “low injury/low shock” (injury severity score < 15, base excess ≥ −6 mEq/L) and “high injury/high shock” (injury severity score ≥ 15, base excess < −6 mEq/L) was used to treat endothelial cells. Compared with low injury/low shock plasma, high injury/high shock induced greater cytosolic Ca^2+^ increase, inducing post-injury endotheliopathy. Furthermore, increased cellular calcium influx has been hypothesized to contribute to post-trauma hypocalcemia.

#### 4.4.4. RhoA GTPase Activation

Increased activation of RhoA GTPase following trauma causes breakdown of endothelial tight and adherens junctions, further enhancing endothelial permeability and contributing to organ dysfunction [58]. Plasma from severely injured patients was added to human umbilical vein endothelial cells. Rho and Rac activity were determined using a G-LISA assay. Endothelial permeability significantly increased with plasma from patients with both severe injury and shock contributing most to this increased permeability [57]. Additionally, incubation with injury + shock plasma resulted in higher RhoA activation (*p* = 0.002) and a trend toward decreased Rac1 activation (*p* = 0.07) compared to the minimally injured control.

Experimental models of trauma suggest that inhibition of RhoA activity in lungs treated with both mesenchymal stem cell-derived extracellular vesicles and mesenchymal stem cells is associated with a decrease in pulmonary cell permeability and lung injury [58].

#### 4.4.5. Inflammatory Mediators

Trauma initiates a systemic inflammatory response characterized by the release of proinflammatory cytokines, complement activation, and neutrophil recruitment. This inflammatory milieu exacerbates endothelial injury and dysfunction [59,60]. This may result in single and multiple organ failure (acute kidney injury, acute respiratory distress syndrome, hepatic dysfunction) and a higher susceptibility to infection. Cellular disruption by trauma releases mitochondrial “damage”-associated molecular patterns (DAMPs) with evolutionarily conserved similarities to bacterial infections. Microbial pathogen-associated molecular patterns (PAMPs) into the circulation. These activate human polymorphonuclear neutrophils (PMNs), leading to PMN migration and degranulation in vitro and in vivo. Circulating MTDs can elicit neutrophil-mediated organ injury [61]. Although a relation has been reported between severity of trauma and cytokine plasma levels, at present their measurement seems of poor prognostic value.

#### 4.4.6. Extracellular Vesicles (EVs)

Extracellular vesicles (EVs) are cellular vesicles ≤1 μm in size that contain membrane fragments, intracellular organelles, exosomes, and associated cargo molecules. They are released from cells undergoing apoptosis or active microvesiculation and contribute to endothelial dysfunction and coagulopathy during acute injury through distinct yet interconnected mechanisms [62]. First, membrane EVs often express anionic phospholipids, which serve as crucial cofactors in the initiation and propagation of coagulation. Second, EVs derived from peripheral blood samples of mice with traumatic brain injury (TBI) induce vasoconstriction both in vivo and in vitro, leading to tissue ischemia [63]. This vasoconstrictive effect appears to be intrinsic to the structure of EVs themselves, as neither the protein nor the lipid fractions extracted from EVs individually induce vasoconstriction. Third, EVs carry bioactive factors originating from parent cells or acquired from plasma, which can activate or damage endothelial cells (ECs). These EVs interact with endothelial cells, promoting inflammation and disrupting endothelial barriers [64].

#### 4.4.7. Metabolic Dysregulation

Trauma alters the metabolic profile of endothelial cells, affecting glucose and fatty acid metabolism. This metabolic shift impairs endothelial function and contributes to the pathophysiology of EoT.

Understanding these intricate mechanisms is fundamental for advancing targeted therapies aimed at alleviating endothelial dysfunction and enhancing outcomes for trauma patients. Continued research is crucial to elucidate the interplay and roles of these pathways in the pathogenesis and evolution of endotheliopathy of trauma.

## 5. Therapeutic Evolution in Trauma-Induced Coagulopathy

The treatment of TIC has developed in parallel with understanding and advancements in diagnostic and therapeutic techniques, focusing on addressing the numerous factors involved in its pathophysiology. This section will examine the treatment options, beginning with traditional and less refined methods and moving towards more targeted approaches guided by diagnostics.(Table 1)

### 5.1. Tranexamic Acid: The Cornerstone

Tranexamic acid (TXA) is extensively studied as an antifibrinolytic agent in trauma care. It acts by inhibiting plasmin, the enzyme responsible for fibrinolysis, through competitive and non-competitive mechanisms. The timing of TXA administration can differentially impact patient outcomes. Moreover, concerns have been raised regarding the potential for TXA to tip the balance between thrombogenesis and thrombolysis, leading to an increased risk of thromboembolic events, such as venous thromboembolism (VTE) or stroke [65]. Known hypersensitivity and subarachnoid hemorrhage are absolute contraindications to the use of tranexamic acid. The latter is due to the increased risk of cerebral edema and ischemia [66]. Side effects such as gastrointestinal disturbance, allergic skin reaction, and visual disturbance are relatively frequent, and acute cortical necrosis and seizures have been reported in particular at high concentrations [67]. Initially, TXA was shown to reduce overall mortality in trauma patients when administered promptly after injury, as evidenced by the CRASH-2 study [68]. Subsequent research highlighted that delaying TXA administration beyond 3 h post-injury increased the risks of venous thromboembolism and hemorrhage-related death [69]. One possible explanation may be the opposite effect of TXA on tPA- vs. uPA-mediated fibrinolysis due to different time-related expression of these plasminogen activators: According to TXA-induced conformation change, plasminogen reduces tPA-mediated activation, whereas it seems to accelerate uPA-mediated activation. International guidelines support the hypothesis that TXA should be administered early after a traumatic event [68]; however, recent meta-analyses have indicated that TXA does not significantly increase the incidence of thrombotic events across various trauma and hemorrhage scenarios. Patients with a fibrinolytic shutdown phenotype, more common in severe trauma cases, may have heightened susceptibility to thromboembolic complications and organ failure with TXA treatment. Consequently, guidelines recommend early TXA administration, ideally within 3 h of injury, especially in settings where TEG can promptly identify fibrinolytic status and guide subsequent TXA dosing.

Current protocols advocate for an initial TXA bolus of 1 g infused over 10 min at the trauma scene, followed by a continuous 8 h infusion to optimize therapeutic benefits.

### 5.2. The Goal-Direct Transfusion

Massive blood transfusion is a cornerstone in managing hemorrhagic shock in trauma patients, favoring improved tissue perfusion and enhanced oxygen delivery. However, despite their benefits, transfusions can exacerbate or trigger TIC. Protocols for massive transfusion vary globally, with ratios of plasma to packed red blood cells (RBCs) ranging from 1:1 to 1:10. Clinical trials such as PROMMTT, PROPPR, and COMBAT have demonstrated that adopting a balanced transfusion protocol with equal parts plasma, RBCs, and platelets (1:1:1 ratio) can enhance survival outcomes for patients at high risk of developing TIC [70].

In the PROMMTT study, early administration of plasma was associated with reduced 24-h and 30-day mortality rates compared to delayed plasma transfusion or lower plasma/RBC ratios. This underscores the critical role of timely plasma infusion in improving outcomes in trauma patients [71].

The PROPPR trial, while not showing differences in overall survival at 24 h and 30 days between the 1:1:1 and 1:1:2 ratios of plasma to platelets to RBCs, indicated that the 1:1:1 ratio group had a lower likelihood of death due to hemorrhage. This highlights the potential benefit of a balanced transfusion approach in mitigating bleeding-related mortality [72].

### 5.3. TEG-Guided Hemostatic Transfusion Strategies

The utilization of TEG in guiding transfusion decisions has shown promising outcomes, including reduced transfusion volumes, improved hemostatic balance, and potentially lower rates of complications such as thromboembolism. The rapid and straightforward approach of TEG could potentially predict the necessity of blood transfusions in severely injured trauma patients. Moving forward, resuscitation protocols informed by TEG findings may emerge as the preferred method. A prospective randomized controlled trial conducted by Gonzalez et al. in 2016 demonstrated reduced mortality among patients transfused based on TEG results compared to those transfused based on conventional laboratory testing (19.6% vs. 36.4%, respectively) [73].

TEG assesses multiple parameters, including reaction time (R-time), clot kinetics (K-time and angle), maximum amplitude (MA), and clot lysis (LY30), each guiding specific decisions regarding transfusion therapy (Figure 3) [74].

R-time: A prolonged R-time suggests the requirement for plasma transfusion. For example, an R-time > 4.45 min indicates the need for fresh frozen plasma (FFP) administration to address coagulopathy.Angle (α): A decreased angle indicates the need for fibrinogen supplementation. An angle < 67 degrees signals the potential need for fibrinogen concentrates or cryoprecipitate infusion.Maximum Amplitude (MA): A reduced MA indicates platelet dysfunction or deficiency. An MA < 60 mm indicates the necessity for platelet transfusion.LY30: Elevated LY30 indicates hyperfibrinolysis, which may require antifibrinolytic therapy like tranexamic acid (TXA). An LY30 > 4.55% suggests the potential benefit of antifibrinolytics.

This strategy has been shown to enhance survival rates and minimize the quantity of blood products administered, supported by research demonstrating strong associations between TEG measurements and patient outcomes in trauma settings [75,76,77].

The threshold values for the most accurate identification of ATC and prediction of massive transfusion (MT) using rotational thromboelastometry (ROTEM) assays are still debated. In the paper by Hagemo et al. [78], the ROTEM CA5 value measured on arrival is a valid marker for ATC and predicts massive transfusion requirements. An EXTEM CA5 threshold value of ≤40 mm has a detection rate of 72.7%, whereas a FIBTEM CA threshold value of ≤9 mm detects MT requirements in 77.5% of cases. In Figure 4 is reported one of the proposed ROTEM-based schemes of treatment.

According to these cut-off values, a recent paper showed that in 51% of patients who had abnormal results at ROTEM in the first 4 h after trauma, the need for red blood cells, frozen plasma, platelet units, or fibrinogen coventrates was 2 to 4 times higher than in those without ROTEM abnormalities [79].

### 5.4. Improving Trauma Care: Fibrinogen Concentrates

Fibrinogen is crucial for hemostasis early after trauma, and low levels following severe injury correlate with impaired clotting, severe bleeding, and poorer outcomes. The timely administration of concentrated fibrinogen has been shown to mitigate TIC and reduce the need for blood transfusions, as evidenced by the randomized controlled FiiRST trial [80]. This trial demonstrated that maintaining high fibrinogen levels through early infusion of concentrated fibrinogen lowers TIC-related complications. Current guidelines recommend fibrinogen replacement when levels drop below 1.5 g/L during significant bleeding, typically achieved using cryoprecipitate derived from fresh frozen plasma.

### 5.5. Pharmacological Interventions

Hemostatic drugs used as adjuncts for managing severe coagulopathy in bleeding patients include recombinant factor VIIa, prothrombin complex concentrate, antifibrinolytic agents, and desmopressin. Recombinant factor VIIa is reserved for life-saving situations due to its role in binding tissue factor exposed during trauma, thereby promoting clot formation at the injury site [81].

Prothrombin complex concentrate (PCC) is enriched in factors II, VII, IX, and X and is commonly used to counteract warfarin-induced anticoagulation due to its high content of vitamin K-dependent clotting factors [82]. PCC showed promising results in managing TIC in select trauma centers, although further clinical studies are required to establish effective therapeutic effects.

Fresh-frozen plasma (FFP) is routinely used to correct coagulopathy in trauma patients by providing volume and coagulation factors support [83]. However, its use is limited by the need for cross-matching and thawing before administration, as well as delayed reversal of coagulopathy [84]. Tissue hypoperfusion early in trauma contributes significantly to coagulopathy, exacerbated by large-volume crystalloid resuscitation, which further dilutes clotting factors [85]. FFP has been recommended for its dual role in providing volume and coagulation support, although logistical challenges persist [29]. Recent interest in PCC as an alternative to FFP may mitigate early coagulopathic effects of fluid resuscitation in trauma patients. However, concerns about thromboembolic risks and higher costs compared to FFP remain significant considerations.

### 5.6. Calcium Chloride

Hypocalcemia in patients requiring massive transfusions may be detrimental, because Ca^2+^ plays a crucial role in normal coagulation. Ca^2+^ is a cofactor in the activation of factors II, VII, IX, and X, along with protein C and protein S of the coagulation cascade, actually factor IV of the coagulative cascade. Moreover, it contributes to platelet adhesion at the site of vessel injury. Hypocalcemia during the first 24 h can predict mortality and the need for multiple transfusions better than the lowest fibrinogen concentrations, acidosis, and the lowest platelet counts. To correct hypocalcemia, calcium chloride is preferred to calcium gluconate, as 10% calcium chloride contains 270 mg of elemental calcium per 10 mL, whereas 10% calcium gluconate contains 90 mg of elemental calcium per 10 mL [71].

## 6. Future Perspectives

As previously reported, multiple associated pathways, including iatrogenic factors, are involved in the pathophysiology of TIC. Replacement of coagulation factors using blood components as well as counteracting enhanced fibrinolysis with tranexamic acid in association with the whole strategy of damage control may contribute to decreasing clinical effects of TIC. Nevertheless, tailored treatment of different mechanisms involved in TIC may offer further therapeutic options.

Glycocalyx shedding plays a significant role in the development of EoT. Therapeutic approaches targeting sheddases responsible for this process or upstream activation of leukocytes that produce these sheddases have been recently studied [86]. Etanercept, angiopoietin-1, hydrocortisone, and heparin may modulate EGL shedding in different clinical conditions, although the benefits of these strategies have not been specifically demonstrated in trauma [87,88,89]. Patients treated with doxycycline, an MMP inhibitor, showed reduced glycocalyx shedding induced by inflammatory and oxidative stress during cardiopulmonary bypass. Syndecan-1 provides structural support to the glycocalyx, while its attached glycosaminoglycans (GAGs), particularly heparan sulfate, are prominently shed in response to trauma and hemorrhagic shock (Table 2). Heparan sulfates with a rare 3-O-sulfate modification possess both anticoagulant and anti-inflammatory properties by binding tightly to antithrombin (AT) and sequestering circulating damage-associated molecular patterns (DAMPs). Recent preclinical studies have shown that a synthetic 3-O-sulfated heparan sulfate, dekaparin, exhibits similar anti-inflammatory and organ-protective effects as plasma in a mouse model of trauma and hemorrhagic shock. Interaction between AT and 3-O-sulfated heparan sulfate-containing HSPGs (3-OS-HSPG) both disrupts coagulation at the endothelial surface and elicits anti-inflammatory effects through induction of prostacyclin synthesis and inhibition of NFκB activation [90]. These findings underscore the biological significance of this unique heparan sulfate molecule and suggest that restoring shed glycocalyx components could represent a promising future therapeutic development.

After trauma, von Willebrand factor (VWF)-mediated extracellular vesicle (EV)-induced vascular activity can be inhibited by exogenous ADAMTS-13, a VWF antibody, or recombinant A2 protein. These interventions target the A1 domain exposed on hyperadhesive VWF in rodent models of trauma, hemorrhagic shock, or traumatic brain injury (TBI) [91]. These findings align with clinical observations in which elevated plasma VWF levels, reduced ADAMTS-13 activity, or an imbalance in the VWF-to-ADAMTS-13 ratio correlate with endotheliopathy, coagulopathy, and adverse outcomes in severely injured patients [92,93,94,95]. Restoring ADAMTS-13 levels could be beneficial. In a rodent model of renal ischemia/reperfusion, Zhou et al. demonstrated that recombinant human ADAMTS-13 (rhADAMTS-13) reduced inflammation and improved endothelial cell function [96]. Similarly, in a mouse model of TBI, Wu et al. found that rhADAMTS-13 effectively mitigated cerebral vascular leakage and coagulopathy and enhanced neurological outcomes and survival [97]. In another experimental model of trauma-induced shock, rats were randomized to receive crystalloids, crystalloids supplemented with rhADAMTS13, or plasma transfusion. Both plasma transfusion and rhADAMTS13 were associated with a reduction in pulmonary endothelial permeability and organ injury when compared with resuscitation with crystalloids, but only rhADAMTS13 resulted in significant improvement of a trauma-induced decline in ADAMTS13 levels [98].

After trauma and under pathological conditions, VWF multimers are released, which activate platelets and endothelial cells (ECs) to secrete procoagulant and proinflammatory EVs. These EVs, bound to VWF, induce vascular hyperpermeability, a critical aspect of endotheliopathy. EVs isolated from severely injured patients injected into naive mice induced a significant increase in endothelial cell injury, as evidenced by syndecan-1 shedding, as well as in coagulopathy and organ injury, compared to mice receiving EVs from minimally injured patients [99,100].

Lactadherin, also known as milk fat globule-epidermal growth factor 8 (MFG-E8), facilitates the removal of pathological EVs by binding them to macrophages for phagocytosis. Lactadherin contains an integrin-binding arginylglycylaspartic acid sequence and two C-terminal domains that strongly bind to phosphatidyl serine. Research by Zhou et al. has demonstrated that lactadherin enhances the clearance of EVs, reduces coagulopathy, prevents vascular permeability, and improves neurological outcomes and survival in mice with severe TBI [101]. Given the extensive tissue damage and ischemia associated with trauma, it is plausible that severely injured patients have elevated levels of circulating EVs compared to those with isolated TBI. Therefore, lactadherin may potentially mitigate endotheliopathy and coagulopathy following trauma and hemorrhage, although clinical studies to confirm this are currently lacking.

Mesenchymal stem cells (MSCs), derived from bone marrow stromal progenitor cells, have been extensively researched both in laboratory settings and clinical trials for their potential therapeutic benefits following traumatic events. They promote angiogenesis to aid in endothelial repair and have demonstrated efficacy in reducing endothelial permeability in models of hemorrhagic shock [102]. Furthermore, studies have shown that EVs derived from MSCs offer similar protective effects as MSCs themselves, indicating that MSC EVs could serve as a viable cell-free therapeutic option following trauma [62]. However, delineating the precise balance between the beneficial and detrimental effects of EVs remains challenging, as their impact can vary based on the source cells, target cells, and the specific cargo they carry.

## 7. Conclusions

In conclusion, EoT results from a complex relation involving the glycocalyx, VWF, and platelets that impair coagulation and endothelial cell function and activate inflammation. Mechanisms underlying endothelial injury and the detrimental effects of EoT, alongside potential therapeutic future options, have been reported. Current therapeutic practice is predominantly based on the administration of plasma and other blood products to reverse volume losses and stop bleeding. Emerging treatment addressed to EC dysfunction showed promising results in animal models. The future challenges in trauma research will be the transferal of tailored treatment in humans and in clinical practice.

## Figures and Tables

**Figure 1 diagnostics-15-01435-f001:**
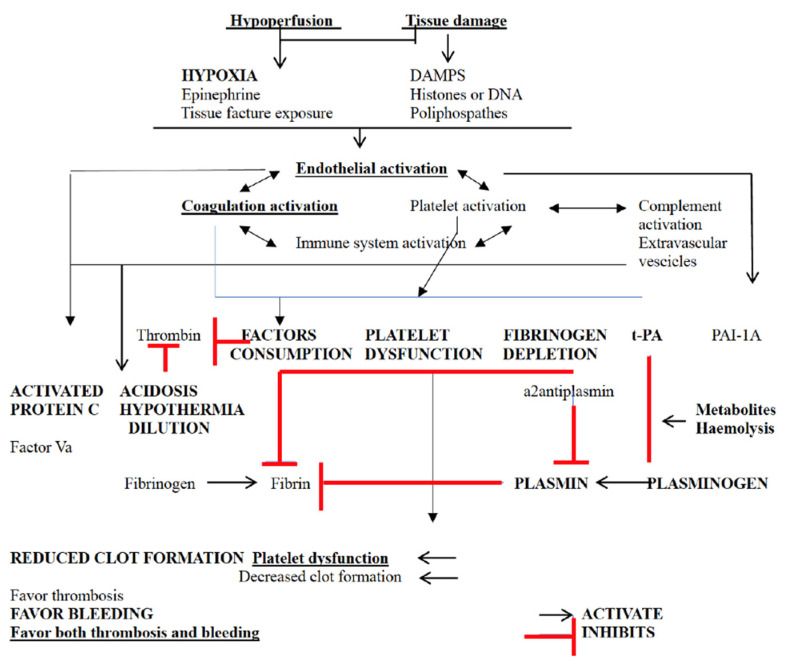
Pathophysiology of trauma-induced coagulopathy.

**Figure 2 diagnostics-15-01435-f002:**
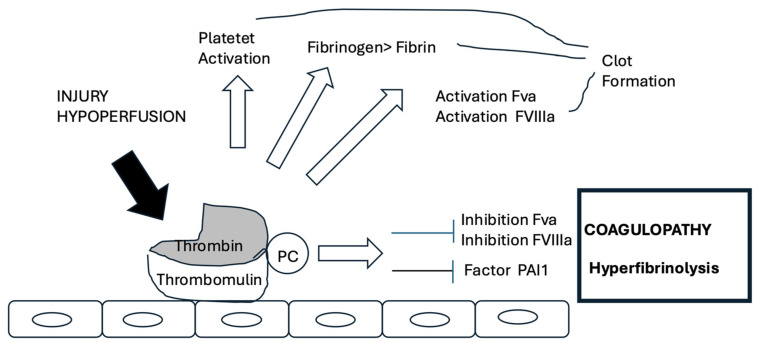
Activation of thrombodulin and TIC.

**Figure 3 diagnostics-15-01435-f003:**
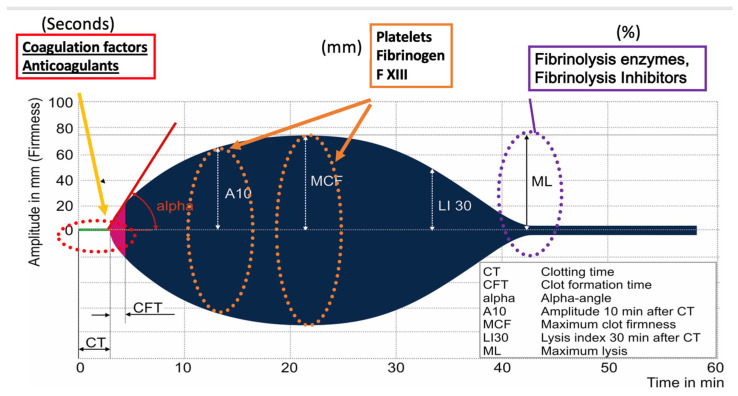
ROTEM curve. Derived variables and their relationship with hemostatic changes are reported.

**Figure 4 diagnostics-15-01435-f004:**
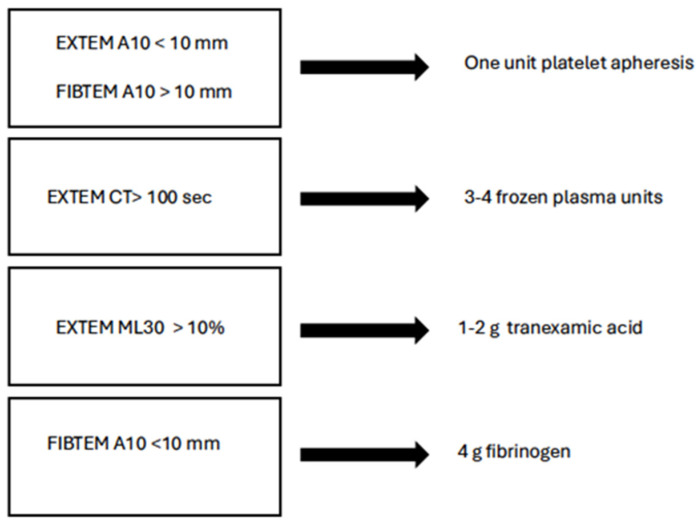
ROTEM based correction of coagulative abnormalities (modified from [76]).

**Table 1 diagnostics-15-01435-t001:** Conventional treatments in trauma-induced coagulopathy/bleeding.

Tranexamic acid	Inhibits fibrinolysis through competitive and non-competitive plasmin inhibition
Goal-Direct transfusion	Balanced transfusion protocol with equal parts plasma, RBCs, and platelets (1:1:1 ratio) can enhance survival
Fibrinogen	Guidelines recommend fibrinogen replacement when levels drop below 1.5 g/L during significant bleeding
Calcium chloride	Restoration of normal calcium levels since calcium have an essential role in the formation and stabilization of fibrin polymerization sites and on platelet function

**Table 2 diagnostics-15-01435-t002:** Innovative treatments of endotheliopathy.

Etanercept angiopoietin-1	Modulation of EGL shedding
Doxycylcyne (MMP inhibitor)	Reduction of glycocalyx shedding
Dekaparin (synthetic 3.0 heparan sulfate)	Modulation of coagulation factors activation, anti-inflammatory effects on endothelial surface
Recombinant-human ADAMTS 13	Decreased coagulopathy and endothelial permeability and induced organ damage
Lactadherin	Facilitate removal of extracellular vesicles
Mesenchymal stem cells	Promotion of angiogenesis and decrease in endothelial permeability

## Data Availability

Not applicable.

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
