# Peer review of "Trauma-Induced Coagulopathy: A Review of Specific Molecular Mechanisms"

_diagnostics, 2025, doi:10.3390/diagnostics15111435_

Round 1

Reviewer 1 Report

Comments and Suggestions for Authors

Dear authors,

Thank you for the opportunity to review your manuscript on traumatic coagulopathy. I read it with great interest as the topic is a "hot" one in the critical care community, and a comprehensive summary of current evidence is surely helpful for students, researchers, and clinicians. I also think that your text is already well written, and the most important subtopics have been covered. However, if you want to publish your text in a peer-reviewed scientific journal rather than as a book chapter, I strongly advise you to also provide a Methods section. I get it - your review is not a systematic review or even a scoping review but rather a narrative one. However, you still came up with your literature and citations according to a plan. Provide your applied methods (PICO, search strategy, databases, in- and exclusion criteria, timeframe, type of literature, etc.) in a transparent way. This would improve your manuscript and distinguish it from a book chapter.

Comments on the Quality of English Language

Generally, the used English style and grammar seem fine. However, some finetuning could still be done throughout the manuscript, preferrably by a native speaker.

Author Response

Thank you for the opportunity to review your manuscript on traumatic coagulopathy. I read it with great interest as the topic is a "hot" one in the critical care community, and a comprehensive summary of current evidence is surely helpful for students, researchers, and clinicians. I also think that your text is already well written, and the most important subtopics have been covered. However, if you want to publish your text in a peer-reviewed scientific journal rather than as a book chapter, I strongly advise you to also provide a Methods section. I get it - your review is not a systematic review or even a scoping review but rather a narrative one. However, you still came up with your literature and citations according to a plan. Provide your applied methods (PICO, search strategy, databases, in- and exclusion criteria, timeframe, type of literature, etc.) in a transparent way. This would improve your manuscript and distinguish it from a book chapter.

A method section was added to the text

Reviewer 2 Report

Comments and Suggestions for Authors

The manuscript "Trauma Induced Coagulopathy: A Review of Specific Molecular Mechanisms" presents a basic review of trauma-induced coagulopathy but lacks the depth and scientific rigor expected for publication.

major comments:

  1. The text is relatively basic and does not explore molecular mechanisms in sufficient detail.
  2. There is a notable absence of critical discussion regarding emerging findings in the field, and the review does not significantly contribute to advancing knowledge on the topic.
  3. The manuscript is not formatted according to the journal’s author guidelines, which raises concerns about the overall preparation and adherence to submission standards.
  4. The figure provided is almost unreadable, making it difficult to interpret its relevance or value to the discussion.
  5.  only one figure is included, which is insufficient to illustrate the complex mechanisms involved in this process

Additionally, several other issues were noted:

1. The introduction and discussion lack depth and fail to integrate key recent findings.

2. The text contains typographical errors, unclear phrasing, and inconsistencies in citations.

3. The structure is disorganized, with transitions between sections and redundant information.

4. The references list contains formatting inconsistencies

Overall, the manuscript requires substantial revision, including an in-depth discussion of molecular mechanisms, clearer scientific explanations, and better adherence to formatting and submission guidelines. The figure should be improved for readability, and additional graphical representations should be incorporated to better illustrate key concepts.

Comments on the Quality of English Language

English language and grammar needs revision.

Author Response

  1. The text is relatively basic and does not explore molecular mechanisms in sufficient detail. The text has been enriched and we hope it may satisfy your suggestions
  2. There is a notable absence of critical discussion regarding emerging findings in the field, and the review does not significantly contribute to advancing knowledge on the topic. Discussion have been reorganized
  3. The manuscript is not formatted according to the journal’s author guidelines, which raises concerns about the overall preparation and adherence to submission standards.  The text has been deeply revised
  4. The figure provided is almost unreadable, making it difficult to interpret its relevance or value to the discussion.   The figure was modified
  5.  only one figure is included, which is insufficient to illustrate the complex mechanisms involved in this process. Other figures were added as required

Additionally, several other issues were noted:

  1. The introduction and discussion lack depth and fail to integrate key recent findings. Introduction has been impoved
  2. The text contains typographical errors, unclear phrasing, and inconsistencies in citations. References have been carefully checked
  3. The structure is disorganized, with transitions between sections and redundant information. The text has been reorganized
  4. The references list contains formatting inconsistencies. References were revised

Overall, the manuscript requires substantial revision, including an in-depth discussion of molecular mechanisms, clearer scientific explanations, and better adherence to formatting and submission guidelines. The figure should be improved for readability, and additional graphical representations should be incorporated to better illustrate key concepts. We tried to improve any aspect reported

Reviewer 3 Report

Comments and Suggestions for Authors

It is a quite interesting manuscript. The topic of this manuscript falls within the scope of Diagnostics Journal.

The authors have provided a comprehensive overview of the pathophysiology of trauma-induced coagulopathy (TIC) and explored both established and emerging management strategies. The literature review is thorough, incorporating relevant studies from PubMed, Scopus, and Web of Science with well-defined search terms. The focus on molecular mechanisms adds depth to the discussion, offering valuable insights into the underlying processes of TIC.

The methodology for article selection should be added,  a more detailed explanation of inclusion and exclusion criteria would improve transparency ( please add diagram)

Overall, this review is a valuable contribution to the field, and with minor revisions,  it could serve as a significant reference for researchers and clinicians.

Author Response

It is a quite interesting manuscript. The topic of this manuscript falls within the scope of Diagnostics Journal.

The authors have provided a comprehensive overview of the pathophysiology of trauma-induced coagulopathy (TIC) and explored both established and emerging management strategies. The literature review is thorough, incorporating relevant studies from PubMed, Scopus, and Web of Science with well-defined search terms. The focus on molecular mechanisms adds depth to the discussion, offering valuable insights into the underlying processes of TIC.

The methodology for article selection should be added,  a more detailed explanation of inclusion and exclusion criteria would improve transparency ( please add diagram).A method section has been added .

Overall, this review is a valuable contribution to the field, and with minor revisions,  it could serve as a significant reference for researchers and clinicians.

Reviewer 4 Report

Comments and Suggestions for Authors

This review article touches upon various features regarding the pathophysiology, diagnosis, and treatment of trauma-induced coagulopathy. Furthermore, it draws attention to the complexity TIC shows, with challenges in management and also its association with high mortality rates.

 The review clearly emphasizes the different molecular mechanisms, supported by contemporary evidence. Along with that, it relies on a well-done exposition of pathology, from the intricacies presented by endothelial dysfunction, hyperfibrinolysis, and the lethal triad of coagulopathy, coupled with hypothermia and acidosis.

Besides the examples, the authors also present an in-depth review of diagnostic modalities, specifically addressing the advantages of viscoelastic hemostatic assays over traditional coagulopathy tests. Such additional inclusion offers practical usage of things like protocol-driven transfusion and the use of tranexamic acid. It would have done well to also give a critical look at some of the therapies discussed in the article and some drawbacks and controversies associated with them.

Tranexamic acid gets highlighted as a core necessary therapy, but further studies on the potential risk of administration, particularly in patients with shutdown fibrinolysis, must be done. Furthermore, a more integrated discussion about the implications of current clinical trials and funneling therapies would serve nicely to enhance the review. It cleverly bridges the gap between clinical practice and molecular mechanisms; thus, it is set to add insight into future studies that will enhance the outcome of patients.

Author Response

This review article touches upon various features regarding the pathophysiology, diagnosis, and treatment of trauma-induced coagulopathy. Furthermore, it draws attention to the complexity TIC shows, with challenges in management and also its association with high mortality rates.

 The review clearly emphasizes the different molecular mechanisms, supported by contemporary evidence. Along with that, it relies on a well-done exposition of pathology, from the intricacies presented by endothelial dysfunction, hyperfibrinolysis, and the lethal triad of coagulopathy, coupled with hypothermia and acidosis.

Besides the examples, the authors also present an in-depth review of diagnostic modalities, specifically addressing the advantages of viscoelastic hemostatic assays over traditional coagulopathy tests. Such additional inclusion offers practical usage of things like protocol-driven transfusion and the use of tranexamic acid. It would have done well to also give a critical look at some of the therapies discussed in the article and some drawbacks and controversies associated with them.  Your suggestion has been followed

Tranexamic acid gets highlighted as a core necessary therapy, but further studies on the potential risk of administration, particularly in patients with shutdown fibrinolysis, must be done. Furthermore, a more integrated discussion about the implications of current clinical trials and funneling therapies would serve nicely to enhance the review. It cleverly bridges the gap between clinical practice and molecular mechanisms; thus, it is set to add insight into future studies that will enhance the outcome of patients.The risks were reported and discussed

Reviewer 5 Report

Comments and Suggestions for Authors

Capponi et al reviewed the molecular mechanisms of trauma-induced coagulopathy. The manuscript at current stage still have various issues that needs to be resolved.

Comments:

  1. Add more details for line 50-52 on the association between TIC and mortality, e.g. HR or OR values.
  2. Describe more details on the value of ROTEM/TEG in operating rooms, line 90-91. Is there any data showing the effect size of these assays in reducing mortality or reducing unnecessary transfusions?
  3. Add reference for line105 on the aberrant activation of APC in hypoperfused tissues.
  4. Figure 1, the layout is very messed up.
  5. Add reference for line150-158!, Also line161-169.
  6. Line 219, “Wang 2019”, pls correctly insert the reference.

Author Response

  1. Add more details for line 50-52 on the association between TIC and mortality, e.g. HR or OR values. Done
  2. Describe more details on the value of ROTEM/TEG in operating rooms, line 90-91. Is there any data showing the effect size of these assays in reducing mortality or reducing unnecessary transfusions? DONE
  3. Add reference for line105 on the aberrant activation of APC in hypoperfused tissues. Added
  4. Figure 1, the layout is very messed up. We tried to improve figure 1.
  5. Add reference for line150-158!, Also line161-169. Added
  6. Line 219, “Wang 2019”, pls correctly insert the reference. Inserted

Round 2

Reviewer 1 Report

Comments and Suggestions for Authors

Thank you for having addressed my comments.

Author Response

Thank you . The manuscript anyway has been further revised according to comments of other reviewers 

Kind regards

Reviewer 2 Report

Comments and Suggestions for Authors

The manuscript entitled "Trauma Induced Coagulopathy: A Narrative Review of Specific Molecular Mechanisms" although with some improvements, does not meet the standard for a narrative review. The abstract is not in accordance with standard scientific review abstracts. The molecular mechanisms stated in the title is not defined in a substantial manner in the manuscript. Figure 1 is not necessary since very basic and moreover, with a typo graphical error "Hypoytermia" The manuscript is more suitable as a chapter in a students' book or for a journal with lower impact factor. Best regards. 

Comments on the Quality of English Language

English language and style needs polishing.

Author Response

Dear Sir 

   the manuscript has been further and deeply reviewed with the hope that now may be suitable for publication.  Figure 1 was removed . 

Kind regards 

Reviewer 5 Report

Comments and Suggestions for Authors

i think readability is a prerequisite for a manuscript. the current vesion of is just unreadable.

Author Response

The manuscript has been further revised.  I hope that in your opinion it may be now readable . Your suggestions at firsrt revision were related to specific questions that were addressed . Since the body of the text was not deeply modified I had difficulties to understand which changes were needed to satisfy  your comment . 

Kind regards 

Round 3

Reviewer 2 Report

Comments and Suggestions for Authors

The authors have made some attempts to address specific aspects of the manuscript; however, the primary concern remains unaddressed: the paper does not present a significant scientific/practical contribution to the field. Despite the revisions, the manuscript still contains many technical errors. These errors, combined with the lack of substantial new insights and comprehensive review, make it challenging to justify publication.

Comments on the Quality of English Language

English language and style needs refinement.

Author Response

Dear Sir , 

   We tried to follow your previous comments . We would like to understand  what do you need to further improve the paper. I review several papers every year and in my opinion any criticism shold be constructive and suggestions understandable for the authors . Thank you 

Reviewer 5 Report

Comments and Suggestions for Authors

The Layout of the manuscript has been notably improved, i have no further comments.

Author Response

Thank you . 

Kind regards